# *Lacerta agilis* and *Zootoca vivipara* Lizards Infested with *Ixodes ricinus* Ticks Preferentially Maintain the Circulation of *Borrelia lusitaniae* and *B. burgdorferi* Sensu Stricto in Poland

**DOI:** 10.3390/ani15213220

**Published:** 2025-11-06

**Authors:** Magdalena Wieczorek, Renata Grochowalska, Bartłomiej Najbar, Bożena Sikora, Jerzy Michalik

**Affiliations:** 1Kepler’s Science Center—Nature Centre, 65-417 Zielona Góra, Poland; m.wieczorek@centrumnaukikeplera.pl; 2Institute of Biological Sciences, University of Zielona Góra, 65-417 Zielona Góra, Poland; b.najbar@wnb.uz.zgora.pl; 3Department of Animal Morphology, Faculty of Biology, Adam Mickiewicz University, 61-614 Poznań, Poland; bozena.sikora@amu.edu.pl

**Keywords:** *Ixodes ricinus*, *Lacerta agilis*, *Zootoca vivipara*, *Borrelia lusitaniae*, Lyme borreliosis, lizards, reptiles

## Abstract

**Simple Summary:**

In Central Europe, lizards are frequent hosts for immature stages of the *Ixodes ricinus* tick, the principal vector of Lyme borreliosis (LB). This is the most common tick-borne disease and is caused by several species of bacteria from the *Borrelia burgdorferi* sensu lato (s.l.) complex. Of the 14 genospecies identified in *I. ricinus*, three account for the majority of LB cases in Europe: *B. afzelii*, *B. garinii* and *B. burgdorferi* sensu stricto. Each of these human-pathogenic species is associated with a particular vertebrate group acting as its natural reservoir. *Borrelia afzelii* depends on rodents, *B. garinii* depends on birds, whereas *B. burgdorferi* s.s. can use both birds and rodents. Certain lizard species are proven reservoir hosts of *B. lusitaniae*, which is implicated as a potential pathogen in humans. This study aimed to evaluate the prevalence of *B. burgdorferi* s.l. in sand lizards and common lizards, as well as in their *I. ricinus* ticks, in suburban areas of western Poland. Our results confirmed that the sand lizard but not the common lizard, can act as a competent reservoir for *B. lusitaniae*. Furthermore, we suggest for the first time that these two lizard species could be another group of reservoir hosts for the human pathogen *B. burgdorferi* s.s., alongside birds and rodents.

**Abstract:**

This study was conducted to assess the involvement of two lizard species: the sand lizard (*Lacerta agilis*) and the common lizard (*Zootoca vivipara*), and their *Ixodes ricinus* ticks, in the circulation spirochetes of the *Borrelia burgdorferi* s.l. complex. Lizards were captured at three study sites in suburban areas of western Poland. Common lizards were less abundant and occurred only at one site. A total of 1129 ticks were collected from 167 sand lizards and 164 individuals from 42 common lizards. Biopsies of the distal part of the lizard tail were taken from 172 animals. All samples that tested positive by real-time PCR underwent subsequent nested PCR targeting the *flaB* gene, followed by sequencing. At least 6.3% of *I. ricinus* ticks (MIR) from *L. agilis*, and 6.1% from *Z. vivipara*, were infected. *Borrelia lusitaniae* was the most prevalent genospecies in *L. agilis*-derived ticks, accounting for 73.2% of all infected samples, followed by *B. burgdorferi* s.s. (23.0%). Conversely, this latter species prevailed (90%) over *B. lusitaniae* (10%) in tick samples from *Z. vivipara.* Therefore, we believe that sand lizards are competent reservoir hosts for *B. lusitaniae*, while the role of *Z. vivipara* for this species is unclear. The high prevalence of *B. burgdorferi* s.s. was also found in infected larval samples (40.7%) and biopsies (60%) of *L. agilis*. Thus, in our opinion, these two lizard species could be another group of reservoir hosts for this human pathogen, along with birds and rodents.

## 1. Introduction

The sand lizard, *Lacerta agilis* Linnaeus, 1758 is widespread throughout Eurasia and has one of the largest ranges of any reptile species, ranging from the western border of France and eastern United Kingdom to Mongolia and north-western China [1]. It is represented by at least ten distinct geographical subspecies belonging to the *L. agilis* complex [2,3]. Sand lizards inhabit a wide range of open or semi-open habitats and occur both in natural and anthropogenic ecosystems characterized by high sunlight exposure, low vegetation, and access to hiding places [4]. They prefer low-dense cover of grass and bushes interspersed with patches of bare ground, and avoid the over-shaded wooded areas. This species is most commonly found in dry grasslands on sandy soils, heathlands, along forest edges, in sparse low pine stands, on grassy or bushy edges of trenches, roadsides, slopes, wastelands, rock rubble, railway and flood embankments [5,6,7].

The common (viviparous) lizard, *Zootoca vivipara* (Lichtenstein, 1823) formerly known as *Lacerta vivipara*, is smaller than *L. agilis*, and its distribution covers nearly the whole of Europe, northern and central Asia and as far as Japan. In consequence, it has the largest and most northerly range of any lizard in the Lacertidae family, including the subarctic regions of Eurasia. Within this distribution, the common lizard has adapted to a cooler, more humid climate [8,9]. It prefers moist, less sunny places, such as peat bogs, wet meadows and forest edges, and avoids dry areas. *Zootoca vivipara* often co-exists simpartically with *L. agilis* lizards [10,11,12]. In Europe, both lizard species are common hosts of *Ixodes ricinus* larvae and nymphs transmitting a wide variety of blood-borne bacterial agents including spirochetes of the *Borrelia burgdorferi* sensu lato (s.l.) complex [13,14]. Currently, this complex comprises at least 23 genospecies (hereafter referred to as species), of which 14 have been reported in European *I. ricinus* ticks [15,16]. At least five of them, *B. afzelii*, *B. garinii*, *B. burgdorferi* sensu stricto (s.s.), *B. spielmanii*, and *B. bavariensis* are considered the causative agents of Lyme borreliosis (LB). The first three predominate in European patients, while the latter two are rarely reported [17,18]. Furthermore, three other spirochaete species, *B. bissettiae*, *B. valaisiana* and *B. lusitaniae*, have been occasionally detected or isolated from human samples and are thought to have pathogenic potential [19].

At least four lizard species: *Lacerta viridis* (Laurenti, 1768), *L. agilis, Z. vivipara* and *Podarcis muralis* (Laurenti, 1768) have been implicated in the maintenance of local cycles of *B. lusitaniae* in Central Europe acting as reservoir hosts for the bacterium [20,21,22,23,24]. This spirochete, which was first detected in Portugal in 1993, occurs in Central and Southeastern Europe, but its prevalence appears to be low and focal. By contrast, in Mediterranean countries such as Portugal, Morocco, Tunisia and Italy, *B. lusitaniae* appears to infect *I. ricinus* ticks more frequently than other spirochaete species [25]. Apart from lizards, this bacterium has sporadically been reported in immature *I. ricinus* ticks collected from birds [26,27] and *Apodemus sylvaticus* mice [28]. Furthermore, de Carvalho et al. [29] isolated *B. lusitaniae* from the same species of mouse. However, these reports did not provide evidence that birds or mice may act as reservoirs for the bacterium. In suitable habitats, lizards can be more important hosts for ticks than rodents or birds. Consequently, their local dominance may negatively impact the spread of spirochaete species other than *B. lusitaniae* [21,30]. Nevertheless, the role of lizards in the circulation of tick-borne pathogens, especially in urban and peri-urban areas, appears to be still underestimated compared to that of mammals and birds [31]. Such areas are highly fragmented environments, composed of a mosaic of patches of different sizes and types of vegetation and land use. However, they might provide suitable conditions for lizards.

The aim of this study was to assess the association between two lizard species (*L. agilis* and *Z. vivipara*) and their larval and nymphal *I. ricinus* ticks in the spreading spirochetes of the *Borrelia burgdorferi* s.l. group in anthropogenically transformed suburban areas of western Poland.

## 2. Material and Methods

### 2.1. Study Sites

The research was conducted from April to September 2017 in three separate study sites (I, II and III) in the Lubusz Province in western Poland (Figure 1). They depict transformed suburban or former agricultural areas that vary in terms of their habitats. Site I: a small village called Olcha that is now part of the city of Zielona Góra (51°52.188′ N, 15°27.122′ E). It has been partially excluded from agricultural use and developed for a new low-rise housing estate. The largest proportion (70%) of the land cover comprised grasslands, arable lands, and mosaics of fresh meadows, with a small *area* covered by riparian forest. Site II: a post-gravel pit area is located within the boundaries of the town of Żary (51°37.690′ N, 15°05.387′ E). Most of the site is covered by young Scots pines, birch and aspen trees, as well as humid meadows. Wetlands with grass and several small seasonal water bodies covered about 30% of the area. Site III: a converted post-sand extraction area is located within the boundaries of the town of Nowa Sól (51°43.850′ N, 15°43.816′ E). It is an exceptionally dry and sunny place dominated by sandy grasslands, which cover over 80% of the slopes. Only 10% of the area was covered by scattered shrubs and trees, and about 5% was a small water body.

### 2.2. Lizard Capture and Tick Collection

In the present study, only lizards infested with ixodid ticks (n = 209) were analyzed. Two species of lizards: the sand lizard *L. agilis* (n = 167) and the common lizard *Z. vivipara* (n = 42) were captured alive by hand. Both species are depicted in Figure 2. The sex (male or female) and age status (adult or juvenile) of the lizards were determined. Each individual was checked for the presence of feeding ticks, which were removed using forceps and stored in 75% ethanol for further analyses. Ticks were identified to the species level using a stereoscope microscope based on morphological criteria according to Siuda [32]. Most of the ticks that fed upon both lizard species were recorded in May and June. Furthermore, a biopsy of up to 1 cm long from the distal part of the lizard’s tail was successfully taken with sterile scissors from 172 animals (131 from *L. agilis* and 41 from *Z. vivipara*). Biopsies were placed in separate vials with 75% ethanol. After examination each animal was released at the site of capture.

### 2.3. DNA Extraction

Genomic DNA from the tails of lizards and ticks was isolated by alkaline hydrolysis, according to previous reference [33]. Ticks were processed individually (60 larvae and 79 nymphs) and in pools (184 larval pools and 89 nymphal pools). A larval pool contained between two and ten individuals, whereas a nymphal pool yielded between two and five. Each pool originated from the same animal. A tick or pooled sample was placed in a 1.5 mL Eppendorf tube and mechanically homogenized using a sterile micropestle. A total of 412 DNA samples (244 from larvae and 168 from nymphs) were obtained from ticks. The obtained lysates were kept at −20 °C.

### 2.4. Screening for Borrelia burgdorferi s.l. DNA

In order to detect *Borrelia* DNA, Real-Time PCR was performed with the primers Bb23Sf and Bb23Sr complementary to a 75 bp fragment of the 23S rRNA gene of *B. burgdorferi* s.l. and with a TaqMan Bb23Sp-FAM probe following the methodology described by Courtney et al. [34]. The reaction was conducted using Real-Time HS-PCR Mastermix Probe (A&A Biotechnology, Gdynia, Poland) following the manufacturer’s instructions. Reaction conditions were as follows: denaturation/polymerase activation at 95 °C for 10 min and 40 cycles of denaturation at 95 °C for 15 s, annealing at 60 °C for 30 s, extension at 60 °C for 30 s. Plasmids pJET1 (A&A Biotechnology, Poland), into which the amplified fragments of the target genes were cloned, were used as positive controls. The negative control consisted of double-distilled water. All real-time DNA amplification reactions were performed using the Mx3005P Real-Time QPCR System (Stratagene, La Jolla, CA, USA) in Department of Tropical Parasitology at the Medical University of Gdańsk.

Positive samples were retested by amplification of the *flaB* gene fragment using two primer sets 132f/905r and 220f/823r [35]. DNA extracted from a tick infected by *B. afzelii* was used to control unspecific detection of *Borrelia* DNA by flagellin gene amplification. Amplification products were separated on 1% agarose gel stained with Midori Green DNA Stain (ABO, Gdańsk, Poland).

### 2.5. Identification of Borrelia Species by Sequencing

PCR-positive products were purified with the clean-up purification (A&A Biotechnology, Gdynia, Poland) and sequenced in both directions by using the same primer pairs (220f and 823r) by firm Macrogen Europe B.V. (Amsterdam, The Netherlands). The obtained sequences (604 bp) were compared with those available in the GenBank databases using BLAST program (US National Institutes of Health, Bethesda, MD, USA) (http://blast.ncbi.nlm.nih.gov/Blast.cgi, accessed on 21 July 2025). Aligned sequences representing *flaB* gene fragments of *Borrelia* strains were examined with MEGA X version 10.0.1 [36]. A total of 96 partial sequences of the *flaB* gene were deposited in GenBank under the following accession numbers: ON086818-ON086836, ON086838-ON086843, ON086845-ON086848; ON086850, ON086852-ON086854, ON086858-ON086860, ON086863-ON086865, ON086881-ON086882-ON086885, ON086887, ON086889, ON086900, ON086902, ON086903, ON086895-ON086909 (*B. lusitaniae*), ON086837, ON086849, ON086851, ON086861, ON086862, ON086866-ON086872-ON086880, ON086888, N086890-N086893, ON086896-ON086899, ON086901, ON086904, ON086910-ON086912 (*B. burgdorferi* s.s.); ON086894, ON086905 (*B. afzelii*), OM970780, OM970781 (*B. miyamotoi*).

### 2.6. Data Analysis

The prevalence of *B. burgdorferi* s.l. infection in ticks was estimated using the Minimum Infection Rate (MIR), i.e., the minimum infected proportion expressed as a percentage: MIR = (p/N) × 100%, where *p* = the number of positive pools and N = the total number of ticks tested. This method assumes that at least one infected tick is present in each positive pool [37]. Differences in the prevalence of *Borrelia* infection in tick samples were evaluated statistically using the 2-tailed chi square test (χ^2^).

## 3. Results

### 3.1. Lizards and Their Ticks

A total of 167 tick-infested sand lizards (*L. agilis*) were captured during our study. They were caught at each of the three different study sites in comparable numbers. In total, 1129 feeding ticks identified as *I. ricinus* (837 larvae and 292 nymphs) were removed from these animals (Table 1). Larvae predominated, accounting for 74% of all ticks, with a ratio of 2.9 larvae to nymphs. Sixty-three animals (37.7%) hosted both larvae and nymphs. Mono-infestation with either larvae or nymphs was recorded in 58 (34.7%) and 46 (27.5%) of the lizards, respectively. On average, one infested animal hosted 6.8 ticks. The number of ticks attached to lizards differed greatly between the study sites. The highest mean intensity of infestation (average number of ticks per tick-infested host) was found at site II, and the lowest at site III (10.8 and 2.8 ticks per animal, respectively).

A total of 42 common lizards (*Z. vivipara*) were captured, all of which were found only at site I. From these individuals, 164 feeding ticks (116 larvae and 48 nymphs) representing *I. ricinus* were collected. A 70% predominance of larvae was recorded, with a ratio of 2.4 larvae to nymphs (Table 1). Ten of the animals (24.4%) were co-infested by both larvae and nymphs. Mono-infestation with either larvae or nymphs was observed in 21 (51.1%) and 10 (24.4%) of the lizards, respectively. The mean intensity reached 3.9 ticks per host, which was comparable to the tick number (5.5 ticks per host) recorded on sand lizards coexisting at the same site.

### 3.2. Borrelia burgdorferi s.l. DNA in Ticks and Lizards

In total, 71 out of 358 tick samples obtained from the sand lizards yielded *Borrelia* DNA. The overall infection prevalence, evaluated as the minimum infection rate (MIR) and calculated for combined tick samples, reached 6.3% (71 of 1129 ticks) (Table 1). The prevalence of *B. burgdorferi* s.l. differed between study sites, ranging from 5.7% to 9.7%. However, these differences were not statistically significant (χ^2^ test, *p* = 0.38). Combined data from all study sites, showed that nymphs were almost five-fold more frequently infected than larvae (15.1% vs. 3.2%, respectively; χ^2^ test, *p* < 0.0001). However, this trend was only recorded at sites II and III (24.2% vs. 1.7%, and 13.6% vs. 7.1%, respectively). In total, 55 (33.0%) out of the 167 lizards hosted at least one *Borrelia* positive tick.

Ten out of 54 tick samples obtained from common lizards were found to be infected with *Borrelia* spp. Thus, at least 6.1% of 164 *I. ricinus* ticks carried spirochetes (Table 1). Infection rates calculated for nymphs and larvae were similar (6.3% and 6.0%, respectively). Eight (9.0%) of the 42 *Z. vivipara* lizards were found to be infested with *B. burgdorferi* s.l. infected ticks. Analysis of biopsies taken from the tails of 131 sand lizards and 41 common lizards revealed that 15 (11.5%) and two (4.9%) of the samples, respectively, yielded *B. burgdorferi* s.l. DNA (Table 2).

### 3.3. Prevalence of B. burgdorferi Sensu Lato Species

*Borrelia lusitaniae* was found to be the most prevalent species in *L. agilis*-derived ticks, accounting for 73.2% (52 out of 71) of all infected samples, followed by *B. burgdorferi* s.s. (23.0%; n = 17), and *B. afzelii* (2.8%; n = 2). The first two species were recorded at all study sites, whereas *B. afzelii* was only detected in two pools, one of which was obtained from larvae and the other from nymphs collected at sites I and II. In PCR-positive nymphal samples *B. lusitaniae* was found to be more common than *B. burgdorferi* s.s. (84.1% vs. 13.6%, respectively). Among the infected larval samples, the two dominant species reached relatively similar prevalences (55.6% and 40.7%, respectively). Furthermore, the DNA of *B. miyamotoi*, a spirochaete belonging to the *Borrelia* relapsing fever group, was found in two nymphs collected from two sand lizards at sites I and III.

In the infected tick samples (n = 10) derived from common lizards, two spirochaete species were identified, with the predominance of *B. burgdorferi* s.s. (90%; n = 9) over *B. lusitaniae* (10%; n = 1). The first species was identified in seven larval and two nymphal samples, whereas the latter was only found in a single pool of nymphs (Table 1).

Of the 15 PCR-positive tail biopsies collected from sand lizards, *B. burgdorferi* s.s. was the most prevalent (60.0%; n = 9), followed by *B. lusitaniae* (33.3%; n = 5) and *B. afzelii* (6.7%; n = 1). *Borrelia lusitaniae* was the only species found in two PCR-positive common lizards (Table 2).

## 4. Discussion

In this study, we provide evidence that two lizard species, *L. agilis* and *Z. vivipara*, along with immature *I. ricinus* ticks, are involved in the circulation of spirochetes from the *B. burgdorferi* s.l. complex in disturbed suburban habitats of western Poland. The sand lizard is the most prevalent and abundant species of lizard in Poland. The presence of the species was identified at all three of the selected locations, while the common lizard was recorded only at one. Both species confirmed their importance as hosts for immature *I. ricinus* ticks, particularly for larvae, which clearly dominated over nymphs, accounting for nearly 74% of the 1293 ticks that were collected. This finding is consistent with the results of our previous two-year study of *L. agilis* in the same area [38]. A similar trend of larval predominance on *L. agilis* and/or on *Z. vivipara* was also reported in southwestern Poland [39,40], Hungary [22] and in the Netherlands [41]. On the other hand, a nymphal dominance over larvae was observed on *L. agilis* in Poland [12,42] and on *L. viridis* in Slovakia [20]. Moreover, in our two-year radiotelemetry study at site I, we removed three times more nymphs (n = 1899) than larvae (n = 628) from 16 monitored sand lizards [43]. According to Dyczko et al. [40], *L. agilis* is a more significant host in maintaining *I. ricinus* populations than *Z. vivipara*. However, in our study, the number of ticks collected from both lizard species which co-occurred only at site I was comparable (5.5 and 3.9 ticks per animal, respectively).

At least 6.3% of *I. ricinus* ticks (calculated as MIR) collected from *L. agilis*, and 6.1% from *Z. vivipara*, were found to be infected with *B. burgdorferi* s.l. It should be emphasized that the overall prevalence of *B. burgdorferi* s.l. found in ticks from the latter species is the highest reported to date. Only 1% of 103 ticks obtained from common lizards in southwestern Poland yielded the bacterium [40]. Apart from the cited Polish report [40], there is no published data on the role of the common lizard in the transmission cycle of *B. burgdorferi* s.l. Studies on sand lizards conducted in Poland showed that the mean infection rates of *B. burgdorferi* s.l. detected in ticks parasitizing these hosts ranged from 4.1% (12/290) [23] to 12% (41/342) [40]. To date, the highest prevalence of the bacterium, 21% (32/152), has been documented by Majláthová et al. [44] in Slovakia. On the other hand, only 0.2% of 1355 ticks from sand lizards examined in a coastal dune ecosystem in the Netherlands carried the bacterium [41].

Despite the finding of comparable prevalences of *B. burgdorferi* s.l. in tick samples obtained from *L. agilis* and *Z. vivipara*, the prevalence of the predominant spirochete species was different. Among PCR-positive tick samples from sand lizards, *B. lusitaniae* proved to be the most prevalent species (73.2%), followed by *B. burgdorferi* s.s. (23.9%). This species prevailed in both infected nymphs (84.1%) and larvae (55.6%), and was present at all study sites. Furthermore, it was detected in 33.3% (5/15) of PCR-positive tail biopsies taken from *L. agilis*. Larval samples obtained from three of these animals also yielded *B. lusitaniae*, implying that they acquired the spirochetes while feeding on them. Therefore, we believe that sand lizards act as competent reservoir hosts for *B. lusitaniae* in suburban habitats where *I. ricinus* populations are present. The involvement of *L. agilis* in the maintenance of *B. lusitanie* was for the first time demonstrated by Richter and Matuschka [21] in Germany. The authors found that all subadult *I. ricinus* ticks that acquired spirochetes from *L. agilis* harbored *B. lusitaniae*. This species distinctly dominated in *Borrelia*-infected ticks collected from *L. agilis* in Slovakia (94%) and Romania (100%) [44]. Studies conducted in Poland also showed that *B. lusitaniae* prevailed in ticks removed from sand lizards, accounting for 66.7% [23] and 88% of all infections [40]. Unexpectedly, *B. lusitaniae* was the dominant species among questing ticks (50%; 21/42) sampled in green areas of Zielona Góra (our site I) [45]. This is the highest infection rate of this species ever recorded in Poland. According to the authors, the high proportion of *B. lusitaniae* indicates a significant expansion of lizards in the city. Therefore, lizards could shape the diversity of *Borrelia* species in ticks inhabiting urban areas. Furthermore, Musilová et al. [24] found that the pathogen was present in 67% of *Borrelia*-infected questing nymphs in the Czech Republic. Of note is that these nymphs were only collected at a site where high numbers of ticks were found on green lizards. Thus, the strong association of this species with lizards may determine its focal distribution in tick populations. The above reports are consistent with the data published by Cirkovic et al. [25], which shows that *B. lusitaniae* is becoming more prevalent in Central Europe and not just in Mediterranean countries.

To date, two strains of *B. lusitaniae* have been isolated from human patients with suspected LB [46] and vasculitis syndrome [47] in Portugal. Furthermore, this spirochete was recently isolated from the blood of a Serbian patient with multiple erythema migrans [48]. These reports implicate the pathogenic potential of *B. lusitaniae* in humans. However, its pathogenicity appears to be limited to some genetic variants [49]. To date, no human cases associated with this spirochete species have been reported in Central European countries.

Our analysis of infected tick samples collected from *Z. vivipara* revealed a clear predominance of *B. burgdorferi* s.s. (90%) over *B. lusitaniae* (10%). To the best of our knowledge, this is the highest rate of the bacterium observed in ticks originating from lizards in Central Europe. Apart from two nymphal samples, the species was identified in larval samples, which were collected from seven animals that were not concurrently infested by nymphs. Therefore, it cannot be excluded that the larvae most likely contracted the infection from the host animals. In this case, it is highly likely that *Z. vivipara* could act as a reservoir host for *B. burgdorferi* s.s., but not for *B. lusitaniae*, which was identified only in one nymphal sample and in two biopsies. This means that *Z. vivipara* exhibits a very low infectivity for the lizard-associated *B. lusitaniae* compared to *L. agilis*, which is its competent reservoir host. This is consistent with the Polish report, in which only one of the 103 ticks parasitizing *Z. vivipara* carried the pathogen [40].

It should be emphasized that we also found very high prevalences of *B. burgdorferi* s.s. in biopsies of sand lizards (60%) and in infected larval samples (40.7%) collected from these hosts. In a study conducted in Hungary, 13% of 31 *Borrelia*-infected ticks obtained from lizards yielded this spirochete species [22]. Similar prevalences of *B. burgdorferi* s.s. in ticks from sand lizards have been documented in Poland with a range from 14.3% [42] to 16.7% [23]. However, Dyczko et al. [40] failed to detect the bacterium in ticks collected from both *Z. vivipara* and *L. agilis* in urban areas of the city of Wrocław in south-western Poland. A low infection rate of 5% (3/60) in ticks from *L. viridis* was reported in the Czech Republic [24].

The human-pathogenic *B. burgdorferi* s.s. is considered a generalist species because it is capable of infecting many different groups of vertebrate species including birds and mammalian hosts, especially rodents [50,51]. Interestingly, this spirochete was recently identified in two chiropterophilic *Ixodes* tick species, which were collected from cave bats captured in Poland and Romania [52]. Therefore, given that various groups of vertebrates and tick vectors can be infected by this generalist species, both *Z. vivipara* and *L. agilis* could locally act as its reservoir hosts. This is important from an epidemiological point of view because *B. burgdorferi* s.s. is responsible for Lyme arthritis (LA) and neurological complications [53]. The data available in Poland show that, in many regions, this is a dominant species among *Borrelia*-infected questing ticks [54,55,56], in contrast to most other European countries [17]. Furthermore, a retrospective study based on data submitted to the Polish National Institute of Public Health revealed that erythema migrans and LA were the most prevalent symptoms of LB between 2015 and 2019 [57]. In our opinion, the high incidence of LA cases could be attributed to the predominance of *B. burgdorferi* s.s. in questing ticks, which acquire the pathogen by feeding on competent reservoir host species.

## 5. Conclusions

Our research provides evidence that two lizard species, *L. agilis* and *Z. vivipara*, along with their immature *I. ricinus* ticks are involved in the circulation of two species from the *B. burgdorferi* s.l. complex, but in different ways. *Lacerta agilis* showed a distinctly higher infectivity for *B. lusitaniae,* as 73% of infected tick samples carried this spirochete, compared to only 10% derived from *Z. vivipara*. Therefore, at our study sites, only the sand lizard served as a competent reservoir host for this pathogen. Further research is needed to determine the reservoir competence of *Z. vivipara* for the lizard-associated *B. lusitaniae*. On the other hand, the unexpectedly high prevalence of *B. burgdorferi* s.s. found in infected ticks collected from *Z. vivipara* (90%) and additionally in biopsies (60%) and larval samples (40.7%) obtained from sand lizards, suggests that these two species could preferentially maintain the circulation of this human-pathogenic spirochete. Of note is that lizard-derived ticks infected with *B. lusitaniae* and *B. burgdorferi* s.s. were found at all study sites, indicating that both spirochete species may be widespread among lizard populations inhabiting suburban areas of western Poland. In such disturbed habitats, lizards could be another group of reservoir hosts for the less specialized *B. burgdorferi* s.s., along with birds and rodents. Based on our data, it can be assumed that this species could potentially be more prevalent in areas inhabited by lizards, but further research is needed to prove this hypothesis. Consequently, the two lizards may influence the diversity of *Borrelia* species in tick populations in urban and suburban areas. In conclusion, our findings highlight the importance of the host element in the ecology of European spirochete species belonging to the *B. burgdorferi* s.l. group.

## Figures and Tables

**Figure 1 animals-15-03220-f001:**
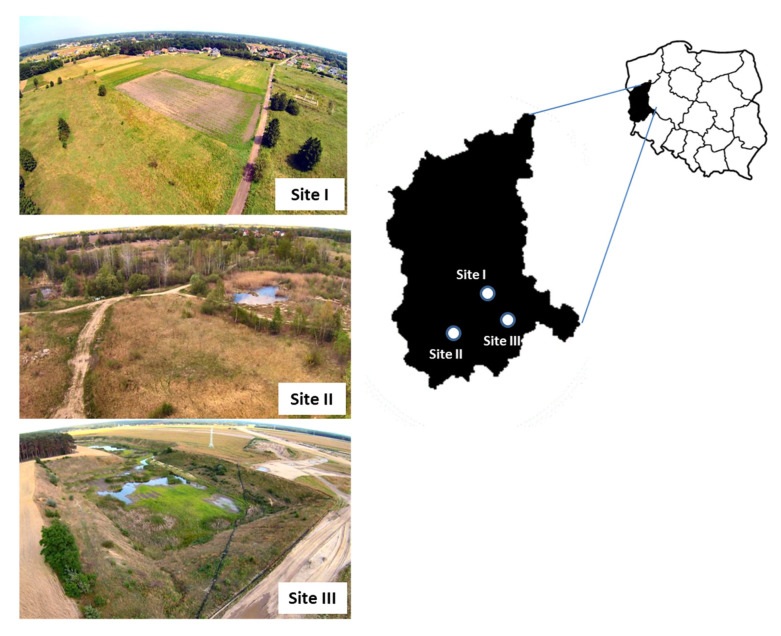
Geographical location of the three collection sites where tick-infested lizards were captured in the Lubusz Province in western Poland. The common lizard was less abundant, being recorded only at site I.

**Figure 2 animals-15-03220-f002:**
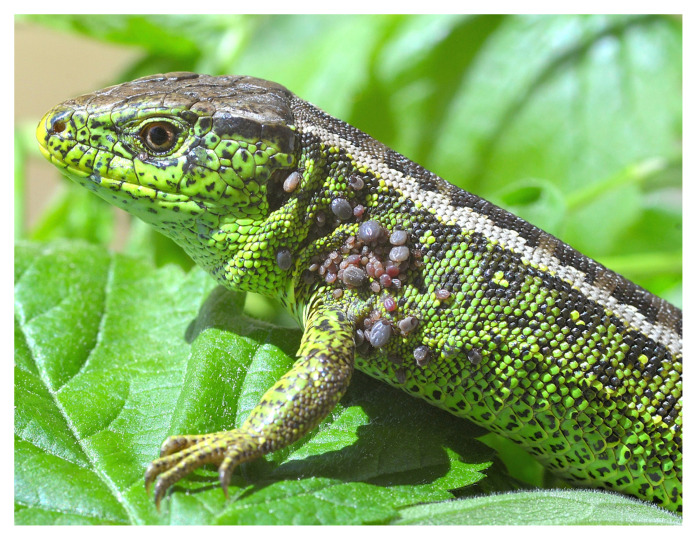
The top photo shows a male of the sand lizard (*Lacerta agilis*), and the bottom photo shows a male of the common lizard (*Zootoca vivipara*). Both lizards are infested with immature *Ixodes ricinus* ticks (photo by B. Najbar).

**Table 1 animals-15-03220-t001:** Prevalence of *Borrelia burgdorferi* s.l. genospecies identified in *Ixodes ricinus* ticks collected from two lizard species: *Lacerta agilis* (LA) and *Zootoca vivipara* (ZV) examined in three separate study sites (I, II and III) in the Lubusz Province in western Poland.

Site	Host (No./MI) ^a^	Tick Stage	No. Collected	No. Isolates Tested/Positive	MIR (%) ^b^	BL ^c^	BB	BA
I (ZG)	LA (55/5.5)	Larvae	171	47/11	6.4	5	6	0
		Nymphs	109	45/6	5.5	2	3	(1) ^d^
		Subtotal	280	92/17	6.1	7	9	1
II (Z)	LA (65/10.8)	Larvae	581	123/10	1.7	8	2	
		Nymphs	124	61/30	24.2	29	1	
		Subtotal	705	184/40	5.7	37	3	
III (NS)	LA (51/2.8)	Larvae	85	41/6	7.1	2	3	1
		Nymphs	59	41/8	13.6	6	2	(1) ^d^
		Subtotal	144	82/14	9.7	8	5	1
TOTAL	LA (167/6.8)	Larvae	837	211/27	3.2	15 (55.6)	11 (40.7)	1 (3.7)
		Nymphs	292	147/44	15.1	37 (84.1)	6 (13.6)	1 (2.3)
		TOTAL	1129	358/71	6.3	52 (73.2)	17 (23.9)	2 (2.8)

I (ZG)	ZV (42/3.9)	Larvae	116	33/7	6.0	0	7	0
		Nymphs	48	21/3	6.3	1	2	0
		TOTAL	164	54/10	6.1	1 (10.0)	9 (90.0)	0

^a^ MI: mean intensity of infestation (average number of ticks per tick-infested host); ^b^ Minimum Infection Rate; ^c^ BL: *B. lusitaniae*; BB: *B. burgdorferi* s.s.; BA: *B. afzelii*; ( ) ^d^ *B. miyamotoi*.

**Table 2 animals-15-03220-t002:** Prevalence of *Borrelia burgdorferi* s.l. species identified in biopsies taken from the distal part of two lizard species’ tails: *Lacerta agilis* (LA) and *Zootoca vivipara* (ZV) examined in three separate study sites (I, II and III) in the Lubusz Province in western Poland.

Site	Host	No. Tested	No. Positive (%)	BL *	BB	BA
I (ZG)	LA	35	0	0	0	0
II (Z)	LA	50	8 (16.0)	4	3	1
III (NS)	LA	46	7 (15.2)	1	6	0
TOTAL	LA	131	15 (11.5)	5 (33.3)	9 (60.0)	1 (6.7)
I (ZG)	ZV	41	2 (4.9)	2	0	0

* BL: *B. lusitaniae*; BB: *B. burgdorferi* s.s.; BA: *B. afzelii*.

## Data Availability

All necessary data are available in the text.

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
