# Peer review of "Lacerta agilis* and *Zootoca vivipara* Lizards Infested with *Ixodes ricinus* Ticks Preferentially Maintain the Circulation of *Borrelia lusitaniae* and *B. burgdorferi* Sensu Stricto in Poland"

_animals, 2025, doi:10.3390/ani15213220_

Round 1
Reviewer 1 Report
Comments and Suggestions for Authors
I have reviewed the manuscript entitled “Sand and Common Lizards (Lacerta agilis and Zootoca vivipara) Infested with Immature Ixodes ricinus Ticks Preferentially Maintain the Circulation of Borrelia lusitaniae and B. burgdorferi Sensu Stricto in Anthropogenic Areas of Western Poland”
The proposed manuscript provides valuable information about two potentially new reservoir hosts for this human pathogen, alongside birds and rodents. The authors provide evidence that two widespread lizard species, L. agilis and Z. vivipara are involved in the circulation of two pathogen species from the B. burgdorferi s.l. complex. These results have significant implications for both the ways in which the pathogen spreads and the role of lizards, given that lizards are often a highly underestimated group in many studies.
Overall, the manuscript is well-written and contains important and valuable results. The language is also of good quality and the text is easy understandable. The abstract is well synthesized and presents a summary of the work done. I encourage the authors to choose key words that are not mentioned in the title.
The results obtained are interesting and valuable, and reflect the research design well. The discussion is comprehensive and contributes to a good understanding of the findings of the study.
I have some minor comments.
It is customary to use the full name, as well as the name of the author and the year of description, when first mentioning a species: Line 52 L. agilis Linnaeus, 1758; Line 60 Zootoca vivipara (Lichtenstein, 1823); Line 76: Lacerta viridis (Laurenti, 1768) and Podarcis muralis (Laurenti, 1768).
Line 54. Provide some citation here.
Line 59 more citation, for example:
Amat F., Lorente G.A. & Carretero M.A. 2003. A preliminary study on thermal ecology, activity times and microhabitat use of Lacerta agilis (Squamata: Lacertidae) in the Pyrenees. Folia Zoology 52 (4): 413–422.
Stumpel A.H.P. 1988. Habitat selection and management of the Sand lizard, Lacerta agilis L., at the Utrechtse Heuvelrug, Central Netherlands. Mertensiella 1: 122–131.
Nemes S., Vogrin M., Hartel T. & Öllerer K. 2006. Habitat selection at the sand lizard (Lacerta agilis): ontogenetic shifts. North-Western Journal of Zoology 2 (1): 17–26.
Čeirāns A. 2007a. Distribution and habitats of the Sand Lizard (Lacerta agilis) in Latvia. Acta Universitatis Latviensis 723: 53–59.
Čeirāns A. 2007b. Microhabitat Characteristics For Reptiles Lacerta agilis, Zootoca vivipara, Anguis fragilis, Natrix natrix, and Vipera berus in Latvia. Russian Journal of Herpetology 14 (3): 172–176.
Heltai B., Sály P., Kovács D. & Kiss I. 2015. Niche segregation of sand lizard (Lacerta agilis) and green lizard (Lacerta viridis) in an urban semi-natural habitat. Amphibia-Reptilia 36: 389–399.
Line 64 more citation, for example:
Horreo, J.L., Pelaez, M.L., Suárez, T., Breedveld, M.C., Heulin, B., Surget-Groba, Y., Oksanen, T.A., Fitze, P.S. (2018): Phylogeography, evolutionary history and effects of glaciations in a species (Zootoca vivipara) inhabiting multiple biogeographic regions. Journal of Biogeography 45(7): 1-12.
Lines 66-69. Can you provide some citations here?
Line 221. B. burgdorferi must be in italic.
Author Response
REVIEWER 1
We would like to thank you for your valuable suggestions and comments. We believe they will help us improve our manuscript. In particular, we consider the additional publications you suggested to be important, as they expand the information on the two species of lizard studied. Our responses to your questions and comments are marked in red font in this revised version of the manuscript. We hope they meet your expectations. Thank you once again for helping us to enhance the substantive value of our manuscript.
Comments 1: I encourage the authors to choose key words that are not mentioned in the title.
Response 1: We have added three terms to the keywords: Lyme borreliosis, lizards, reptiles.
Comments 2: It is customary to use the full name, as well as the name of the author and the year of description, when first mentioning a species: Line 52 L. agilis Linnaeus, 1758; Line 60 Zootoca vivipara (Lichtenstein, 1823); Line 76: Lacerta viridis (Laurenti, 1768) and Podarcis muralis (Laurenti, 1768).
Response 2: It has been corrected.
Comments 3: Line 54. Provide some citation here.
Response 3: The below publication has been added:
- Bischoff, W. Zur Verbreitung und Systematik der Zauneidechse, Lacerta agilis LINNAEUS, 1758. Mertensiella. 1988, 1,11–30.
Comments 4: Line 59 more citation:
Response 4: The bellow three publications have been added:
- Nemes, S.; Vogrin, M.; Hartel, T.; Öllerer, K. Habitat selection at the sand lizard (Lacerta agilis): ontogenetic shifts. North-West. J. Zool. 2006, 2, 17–26.
- Čeirāns, A. Microhabitat Characteristics For Reptiles Lacerta agilis, Zootoca vivipara, Anguis fragilis, Natrix natrix, and Vipera berus in Latvia. Russian Journal of Herpetology. 2007, 14, 172–176.
- Heltai, B.; Sály, P.; Kovács, D.; Kiss, I. Niche segregation of sand lizard (Lacerta agilis) and green lizard (Lacerta viridis) in an urban semi-natural habitat. Amphibia-Reptilia. 2015, 36, 389–399.Nemes S., Vogrin M., Hartel T. & Öllerer K. 2006. Habitat selection at the sand lizard (Lacerta agilis): ontogenetic shifts. North-Western Journal of Zoology 2 (1): 17–26.
Comments 5: Line 64 more citation, for example:
Response 5: The bellow publication has been added:
Horreo, J.L., Pelaez, M.L., Suárez, T., Breedveld, M.C., Heulin, B., Surget-Groba, Y., Oksanen, T.A., Fitze, P.S. Phylogeography, evolutionary history and effects of glaciations in a species (Zootoca vivipara) inhabiting multiple biogeographic regions. J. Biogeogr. 2018, 45, 1-12.
Comments 6: Lines 66-69. Can you provide some citations here?
Response 6: The bellow two publications have been added:
- Mendoza-Roldan JA, Mendoza-Roldan MA, Otranto D. Reptile vector-borne diseases of zoonotic concern. Int J Parasitol Parasites Wildl. 2021, 15:132–42. doi: 10.1016/j.ijppaw.2021.04.007
- Nowak TA, Burke RL, Diuk-Wasser MA, Lin YP. Lizards and the enzootic cycle of Borrelia burgdorferi sensu lato. Mol Microbiol. 2024, 121, 1262-1272.
Comments 7: Line 221. B. burgdorferi must be in italic.
Response 7: It has been corrected
Reviewer 2 Report
Comments and Suggestions for Authors
Wieczorek and colleagues present novel data on the occurrence of Borrelia burgdorferi s.l. in ticks and two species of lacertid lizards, Zootoca vivipara and Lacerta agilis, in one and three periurban areas of Poland, respectively for each of the lizard species investigated. They found differences in the molecular prevalence of this tick-borne pathogen across levels of tick stage (larvae and nymphs) and tail biopsies of lizard hosts. I think the paper is interesting because it illustrates not just the occurrence of this spirokaeta in tick infesting lizards and lizard themselves, but also that prevalence can geographically vary and across levels of the tick stage and lizard species investigated. I have just very few corrections that will be easily addressed by the authors. I think this paper is suitable for Animals.
Line 37: For clarity, change “164 individuals” to “164 ticks”.
Line 66: correct “simpartically”
Line 69: I think a standard term to refer to the species within the B. burgdorferi s.l. complex is “genospecies”. Consider using it throughout the text.
Are the terms “prevalence” and “predominance” the same? Please, consider further clarify it if not, otherwise (if so) just use one term throughout the manuscript for consistency and clarity.
Author Response
REVIEWER 2
We would like to thank you for your valuable suggestions and comments. We believe they will help us improve our manuscript. Our responses to your questions and comments are marked in dark blue font in this revised version of the manuscript. We hope they meet your expectations. Thank you once again for helping us to enhance the substantive value of our manuscript.
Comments 1: Line 66: correct “simpartically”
Response 1: It has been corrected.
Comments 2: Line 69: I think a standard term to refer to the species within the B. burgdorferi s.l. complex is “genospecies”. Consider using it throughout the text.
Response 2: We have amended the sentence relating to the term 'Borrelia (geno)species'. “Currently, this complex comprises at least 23 species genospecies (hereafter referred to as species), of which 14 have been reported in European I. ricinus ticks”.
Comments 3: Are the terms “prevalence” and “predominance” the same? Please, consider further clarify it if not, otherwise (if so) just use one term throughout the manuscript for consistency and clarity.
Response 3: PREVALENCE refers to the proportion of specimens (e.g. lizards) in a tested sample that are infected or infested with a particular pathogen or parasite. It is calculated as a percentage by dividing the number of infected specimens by the total number examined.
PREDOMINANCE (= dominance) means that a particular parasite species or its specific developmental stage (e.g. larvae or nymphs), or a specific pathogen species, is more common (prevails = predominates) within a group of examined individuals (hosts or ticks). Prevalence is a specific measure of frequency expressed as a percentage in a population, whereas predominance is a general (comparative) term for dominance. For example, Borrelia lusitaniae was more common (prevalent, widespread) in nymphs than in larvae.
We don't think it's necessary to explain these two terms in the text.
Reviewer 3 Report
Comments and Suggestions for Authors
The manuscript provides an important contribution to understanding the epidemiological circulation of borreliosis, highlighting the role of lizards as potential reservoir hosts. Together with other studies, these findings emphasize the importance of expanding zoonotic research among reptiles, especially in widespread species living in urban and suburban areas to better understand the parasite and bacterial transmission cycles and their relevance to human health within the One Health concept.
General comments:
- The current title is longer than necessary. A more concise version is recommended
- It would improve the manuscript to include the number of infected versus non-infected lizards, if available. This data would clarify the prevalence and help assess host competence and tick infection dynamics.
- Why only Ixodes ricinus ticks were noted. Is this due to only one species presence in the studied region, habitat preference, or collection bias?
Minor comments:
- Line 33: Uniform presentation of species names. The first mention should include the full genus name (Lacerta agilis)
- Line 94: The species name is not italicized
- Lines 127–128: The sentence “The age (adult female, adult male, or juvenile specimens) and sex of the lizards were determined” should be revised. Suggested version: “The sex (male or female) and age status (adult or juvenile) of the lizards were determined.”
- Line 221: The text states that “these differences were not statistically significant” — please specify which statistical test was used and provide the p-value.
Author Response
REVIEWER 3
We would like to thank you for your valuable suggestions and comments. We believe they will help us improve our manuscript. Our responses to your questions and comments are marked in green font in this revised version of the manuscript. We hope they meet your expectations. Thank you once again for helping us to enhance the substantive value of our manuscript.
Comments 1: The current title is longer than necessary. A more concise version is recommended
Response 1: We agree that this title is rather long. However, we believe that it accurately reflects the study's key findings, it is informative. Therefore, we removed eight words from the title and we kindly ask the reviewer to accept the amended version.
Lacerta agilis and Zootoca vivipara lizards infested with Ixodes ricinus ticks preferentially maintain the circulation of Borrelia lusitaniae and B. burgdorferi sensu stricto in Poland
Comments 2: It would improve the manuscript to include the number of infected versus non-infected lizards, if available. This data would clarify the prevalence and help assess host competence and tick infection dynamics.
Response 2: In the present study, only lizards infested with ticks (n=209) were analyzed and this information was included in Materials and Methods. The results concerning the infection of animals infected with Borrelia spirochetes, based on the analysis of biopsies taken from the tails of 131 sand lizards and 41 common lizards, are presented in Table 2.
Comments 3: Why only Ixodes ricinus ticks were noted. Is this due to only one species presence in the studied region, habitat preference, or collection bias?
Response 3: Available data from Europe shows that lizards are infested with immature stages of Ixodes ricinus. Therefore, only this tick species was found in our study area.
In a recent report by Orlova et al. (2023), apart from Ixodes ricinus, Haemaphysalis punctata, Hyalomma marginatum and Haem. caucasica ticks were found on lizards from the Caucasus and Adjacent Territory (https://doi.org/10.3390/d15091026)
Comments 4: Line 33: Uniform presentation of species names. The first mention should include the full genus name (Lacerta agilis)
Response 4: It has been corrected.
Comments 5: Line 94: The species name is not italicized
Response 5: It has been corrected.
Comments 6: Lines 127–128: The sentence “The age (adult female, adult male, or juvenile specimens) and sex of the lizards were determined” should be revised. Suggested version: “The sex (male or female) and age status (adult or juvenile) of the lizards were determined.”
Response 6: It has been revised.
Comments 7: Line 221: The text states that “these differences were not statistically significant” — please specify which statistical test was used and provide the p-value.
Response 7: the test has been performed χ2 test, P<0.38